# Prevalence of Adverse Childhood Experiences in the First Decade of Life: A Study in the Portuguese Cohort, Generation XXI

**DOI:** 10.3390/ijerph19148344

**Published:** 2022-07-08

**Authors:** Sara Soares, Armine Abrahamyan, Mariana Amorim, Ana Cristina Santos, Sílvia Fraga

**Affiliations:** 1EPI Unit—Instituto de Saúde Pública, Universidade do Porto, Rua das Taipas, n° 135, 4050-600 Porto, Portugal; sara.soares@ispup.up.pt (S.S.); aabrahamyan@ispup.up.pt (A.A.); mariana.amorim@ispup.up.pt (M.A.); acsantos@med.up.pt (A.C.S.); 2Laboratório para a Investigação Integrativa e Translacional em Saúde Populacional (ITR), Universidade do Porto, Rua das Taipas, n° 135, 4050-600 Porto, Portugal

**Keywords:** adverse childhood experiences, violence, childhood

## Abstract

Adverse childhood experiences (ACEs) are a modifiable risk factor for diseases throughout life. This study estimates the prevalence of ACEs in children, addressing associated sociodemographic characteristics and examining the relationship of ACEs with the child’s health and behaviors. We used information on 5295 participants at 10 years old, of the birth cohort Generation XXI, established in Porto, Portugal. Children answered a self-administered questionnaire on ACEs, based on the original ACEs study. Principal component analysis was used to group correlated ACEs, and a score was computed to assess their cumulative effect. Overall, 96.2% of children reported having been exposed to at least one ACE. The most prevalent ACE was a household member shouting, yelling, or screaming at the child (57.7%). Boys were more likely than girls to report “abuse”, “school problems”, and “death/severe disease”. Low parental education, income, and unemployment were associated with an increased risk of “school problems”, “death/severe disease”, and “household dysfunction”. We observed that the dimensions of ACEs could be identified at 10 years of age. A disadvantaged socioeconomic environment was associated with dimensions of ACEs. These data illustrate the natural history of dimensions of ACEs and their potential social patterning.

## 1. Introduction

Adverse childhood experiences (ACEs) are potentially traumatic events occurring in the first 18 years of life [1]. These experiences include psychological, physical, and sexual forms of abuse, as well as household dysfunction (e.g., history of substance abuse, mental illness, violence, and incarceration of a household member) [1,2].

The European status report on preventing child maltreatment documented a high prevalence of ACEs: 9.6% for sexual abuse, 16.3% for physical neglect, 18.4% for emotional neglect, 22.9% for physical abuse, and 29.6% for emotional abuse [3]. More recently, a systematic review of studies published between 1990 and 2015, examining multiple adverse events in national samples from the United States of America, estimated the prevalence of ACEs in school-aged youth ranging from 41% to 97% [4]. Thus, the prevalence of ACEs varies widely across studies and depends on the operational definition of ACEs, the assessment, the recall period, and the contextual environment. Particularly for the Portuguese samples, the results of a study aiming at evaluating the prevalence of adverse experiences in a sample with 75 participants with morbid obesity who were candidates for bariatric surgery showed that 66 (88%) participants had at least one adverse childhood experience, and 51 (68%) referred to at least four adverse experiences [5]. In a sample of 192 participants, 31.8% (*n* = 61) did not report any adverse experience in the first 18 years of life, while 131 (68.2%) participants reported at least one ACE. Of these, 28.1% (*n* = 54) reported one ACE, 24.5% (*n* = 47) reported two to three ACEs, and 15.6% (*n* = 30) reported four or more experiences [6]. Another study in young adults between 18 and 30 years of age showed that a traumatic experience is reported by 24% of the participants, and 13,5% reported having witnessed a traumatic event [7]. Another Portuguese study exploring a relationship between childhood adversity and the risk of incarceration showed that the incarcerated group had significantly more reported adversity, global psychopathology, and global index of risk behaviors than the institutionalized, home, and comparison groups [8].

A growing body of research has been showing the importance of early life experiences to people’s health throughout the life course [1,9,10,11]. In fact, individuals who have experienced such events in childhood or adolescence are more likely to have physical and mental health problems as adults than those who do not experience adversity [1,11,12,13,14]. Lifelong effects of these experiences include increased risk of developing ischemic heart disease, cancer, stroke, diabetes, asthma, or chronic bronchitis later in life, and premature death [1,11,12,15]. Additionally, it is expected that ACEs cluster in children’s lives, and a growing body of research has identified cumulative relations between multiple ACEs and poor physical and mental health later in life [16,17,18]. However, ACEs can also have more immediate or short effects on health already in childhood. Individuals experiencing ACEs are more susceptible to having increased somatic complaints, weight gain and being overweight or obese [19,20], low-average performance at school [21], and behavioral problems [21,22,23] at early ages. Thus, children experiencing early life adversity are at increased risk of poor health later, which may not be revealed until adulthood.

A conceptual framework explaining how childhood exposure to chronic stress, induced by ACEs, leads to changes in the development of nervous, endocrine, and immune systems, resulting in impaired cognitive, social, and emotional functioning and increased physiological damage, has been described [15,24,25]. It is believed that the abnormal stress response consists of disorganization of the neuro–endocrine–immune response, resulting in prolonged cortisol activation and a persistent inflammatory state, with the body failing to compensate after the source of stress is eliminated [24]. Thus, ACEs are preventable risk factors for diseases, which may be prevented [26].

Most of the studies on ACEs were conducted in adult populations, raising problems of recall bias, since data are self-reported and collected retrospectively [27,28,29]. Some of these authors used principal component analysis to derive dimensions for ACEs and find their association with cancer diagnosis [30], physical and mental health, stress, resilience [30], and HPA axis activity [31]. However, all were conducted in smaller samples and when participants were adults. So, it is of major importance to use prospective studies to better understand the occurrence of ACEs and to monitor and assess these experiences already at early ages at the population level [32]. However, some controversy has arisen regarding the validity of the ACEs scale in predicting an individual’s risk of later health problems [33,34,35]. Nevertheless, using the ACEs questionnaires in clinical practice does not inevitably entail “formal” screening but can be used as a strategy to open a conversation about ACEs or a component of trauma-informed care. Thus, ACE questions can be included in history-taking sensitively and safely, which considers the needs of the patient [36].

Therefore, the current study aims to estimate the frequency of ACEs in 10-year-old children from a population-based cohort by examining how the individual items of ACEs group and gather into different dimensions of ACEs and to estimate how these dimensions are associated with sociodemographic characteristics and child characteristics and health-related behaviors.

## 2. Materials and Methods

### 2.1. Study Design and Participants

The study sample consisted of children who participated in Generation XXI, a prospective Portuguese population-based birth cohort. Briefly, of the invited mothers, 91.4% agreed to participate, and their 8647 infants delivered in 2005–2006 in the Porto Metropolitan Area in northern Portugal were enrolled in the cohort [37]. Since then, the entire cohort was invited to attend the first, second, and third follow-up assessments, when children were 4, 7, and 10 years of age (86.3%, 79.6%, and 73.9% participation rates, respectively). Anthropometric measures and blood samples were collected in all study waves, following the same standardized procedures. Data on demographic and socioeconomic characteristics, personal history of disease, and health-related behaviors were collected by trained interviewers through structured questionnaires.

The cohort was approved by the National Data Protection Authority and by the ethics committee of Hospital São João (CES-01/2017). Data confidentiality and protection were guaranteed in all procedures according to the Declaration of Helsinki. Written informed consent was obtained for all participating children, signed by their legal guardians at every study wave.

At the age of 10, all cohort participants were invited to the third follow-up assessment, which could take up to two hours to be completed. The analysis of the present study includes data from all participants with complete information on the ACEs questionnaire, i.e., questionnaires fully completed. Thus, the analyses were based on data from 5295 participants (2598 girls and 2697 boys), and the sample characteristics are presented in Appendix A.

A comparative analysis was conducted between the group of participants with complete information on the ACEs questionnaire and those with incomplete information for the present study. Data indicated that non-participants were more frequently from low-income families (40.1% in the category ≤ EUR 1000 vs. 26.8% in the category > EUR 1000 of family income per month, *p* < 0.001).

### 2.2. Adverse Childhood Experiences (ACEs)

At the 10-year-old assessment, children answered a self-administered questionnaire on ACEs exposure using 15 questions adapted from the ACEs study based on the original ACEs study [1] and ACEs questions adapted from the Child and Adolescent Survey of Experiences: Child Version (CASE) [38]. The information was collected through self-administered questionnaires in a private environment, and the child was helped by a trained interviewer whenever requested. Children reported their lifetime experience of moving from a house, school, or neighborhood against their will; learning problems at school; the death of a family member; injury or serious illness in the family; child hospitalization due to a disease or an accident; parents called to school because the child was in trouble; parental divorce or separation; financial issues in the household; a family member with a drug or alcohol addiction; incarceration of a household member; witnessing parents arguing or fighting; experiencing someone in the household shouting, yelling, or screaming; insulting or humiliating the child; and finally, being hit, kicked, or punched by someone at home. For each item, children could choose “yes” or “no” as to whether the adversity had happened to them or not, respectively.

### 2.3. ACEs Dimensions Derived by Principal Component Analysis

Principal component analysis (PCA) forms linear combinations of the original observed items of ACEs by grouping correlated ACEs, reducing the size of the dataset by identifying the underlying dimensions within the data [39]. The factor loadings are the coefficients defining the linear combinations of ACEs and represent the correlations of each ACEs item with the corresponding dimension. The number of dimensions that best represents the data is chosen based on the scree plot [40], the size of the factor loadings, and the theoretical interpretability of the resulting dimensions. Varimax rotation [39,41] is applied; this redistributes the explained variance for the individual dimensions and provides a simpler structure, increasing the number of larger and smaller loadings.

ACEs items with loadings >0.300 on a dimension were considered to have a strong association with the dimension and were deemed to be the most informative in describing it [42]. Table 1 shows the items of ACEs with factor loadings >0.300 on each dimension, the labels applied to the dimensions, and the amount of the variance unexplained by each item. Thus, ACEs were grouped in five dimensions: abuse, school problems, death/severe disease, life changes, and household dysfunction (Table 1).

To capture the cumulative effects of multiple ACEs, another variable comprising the sum of the different dimensions was composed ranging from 0 (no dimensions) to 5 (five dimensions) (Appendix A).

### 2.4. Sociodemographic Characteristics

Information on sociodemographic characteristics was reported by parents. The monthly disposable household income included salaries and other sources of income, such as financial assistance, rent, monetary allowances, and alimony, for all the household.

Low household income was defined as having equal to or less than EUR 1000 disposable per month, which represented a situation of both parents receiving at least the national minimum wage (EUR 557 before taxes, in 2017 [43]). The intermediate category was defined as between EUR 1001 and EUR 2000 per month, and the highest category was defined as higher than EUR 2000 per month.

Parental education was measured as the number of years of formal schooling completed and classified according to the 2011 International Standard Classification of Education classes [44]. The low educational level corresponded to 9 years or less of formal schooling; intermediate education from 10 to 12 years of formal education; and high education to more than 12 years of formal education.

Parental employment status was represented as a categorical variable with three response options: one parent employed, both parents employed, and both parents unemployed.

Family structure refers to the combination of relatives who comprise a family, and this variable was classified into three categories: none of the parents, one of the parents, or both parents.

### 2.5. Child Health Status and Health-Related Behaviours

The medical diagnosis of any disease was collected by asking parents if the child was ever diagnosed with any disease during their lifetime. Among the medical diagnoses of diseases, asthma is the most common chronic condition in children [45] and also the most frequently reported in Generation XXI, and consequently, it was considered as a separate variable in our analysis.

Body mass index (BMI) was calculated as weight (kg) over squared height (m) and computed as age- and sex-specific BMI standard deviation [46] scores (z-score), according to the World Health Organization (WHO) Child Growth Standards (5–19 years) [46]. BMI z-score categories were defined with the following cut-off points: “underweight, <−2 SD”; “normal weight, −2 ≤ SD ≤ +1”; “overweight, +1 < SD ≤ +2”; and “obese, >+2 SD”.

The daily intake of fruits, soup, and boiled and raw vegetables was collected and was used to compute a variable that estimated the consumption of at least five portions of fruits and vegetables each day (five-a-day fruit and vegetables). This variable followed the recommendations of the WHO for the daily consumption of 400 g of fruits and vegetables and was used as a proxy to measure healthy eating habits [47] and was dichotomized into “less than five” and “five or more portions per day”.

Excess of screen activities was defined as the total time spent on recreational screen time (i.e., television watching, computer and mobile devices use, video gaming), as well as reading, studying, or doing homework (not accounting for school hours), considering weekdays and weekends. It was dichotomized into “yes”, defined as spending more than 480 min (high exposure to screen activities, corresponding to the fifth quintile of the sample distribution) per week on screen activities, considering weekdays and weekends, and “no” if otherwise.

### 2.6. Statistical Analysis

The prevalence of each ACE was estimated (*n*, %) (Table 1). Principal components analysis (PCA) was used to obtain ACEs dimensions (*n* = 5), taking into consideration the shared information across the ACEs. Each child reporting any item included in the different dimensions was included in the specific ACEs dimension. The component loading pattern (the correlation coefficients of the original variables with the components) is presented in Table 1. Bivariate analyses (*n*, % and 95% confidence intervals (CI)) were used to examine the association between family structure, maternal and paternal education, income, history of parental unemployment, children’s sex, consumption of fruits and vegetables, excess of screen activities, any disease diagnosis, asthma diagnosis and BMI, and exposure to each of the 5 dimensions of ACEs (Table 2). Logistic regression models were used to compute odds ratios (OR) and 95%CI for the association between the five principal dimensions of ACEs and sociodemographic characteristics (Table 3) and for the sum of the dimensions and the sociodemographic characteristics (Table 4). Adjusted OR (AOR) and 95%CI were calculated for the association between the five dimensions of ACEs and child health status and behaviors, adjusting for sex and household income (Table 3). AOR and 95%CI were also calculated for the association between the sum of the dimensions and child health status and behaviors, adjusting for sex and household income (Table 4). Analyses were performed using the software Stata^®^ version 15.1 (StataCorp. 2017. Stata Statistical Software: Release 15. College Station, TX, USA: StataCorp LLC).

## 3. Results

Table 1 shows the frequency of the ACEs according to sex and how the different items of ACEs were grouped into the final five dimensions. The most prevalent types of ACEs reported were a household member shouting, yelling, or screaming at the child (57.7%), witnessing parents arguing or fighting (44.4%), the death of someone very close to the child (43.5%), and ever being beaten and hurt at school (42.8%). Having a household member hitting, kicking, or punching the child was reported by 18.5% of children (16.4% of girls and 20.4% of boys). The five dimensions identified and the respective prevalence were: “abuse” (71.5%), “school problems” (63.2%), “death/severe disease” (63.5%), “life changes” (51.0%), and “household dysfunction” (25.6%).

Table 2 shows the association between sociodemographic characteristics and the five dimensions of ACEs. “Abuse” and “life changes” occurred more frequently in households with higher levels of maternal and paternal education. On the contrary, the frequency of “school problems” and “household dysfunction” decreased with lower maternal education levels. “School problems” and “household dysfunction” were more frequent in children living with one parent, or neither parent, and when both parents were ever unemployed (*p* < 0.001). “Death/severe disease” and “life changes” were also more frequent in children living with one parent or neither parent. The same trend was observed for paternal education. Being in the lower category of household income increased the frequency of all dimensions of ACEs.

“Household dysfunction” was more frequently reported by children presenting low consumption of fruits and vegetables, and all dimensions of ACEs were more frequently reported in children with excess screen activities.

Reporting “death/severe disease” increased the frequency of “any disease diagnosis” and “asthma diagnosis”. “Household dysfunction” presented a dose–response association with BMI, with an increased frequency of children reporting it in the higher BMI categories (Table 2).

Lower parental educational level (≤9 years) was associated with an increased risk of “school problems” (OR = 1.61; 95%CI = 1.40–1.86), “death/severe disease” (OR = 1.15; 95%CI = 1.00–1.32), and “household dysfunction” (OR = 1.63; 95%CI = 1.40–1.89). Low household income (≤EUR 1000) increased the likelihood of “school problems” (OR = 1.59; 95%CI = 1.40–1.82), “death/severe disease” (OR = 1.20; 95%CI = 1.05–1.36), “life changes” (OR = 1.72; 95%CI = 1.52–1.95), and “household dysfunction” (OR = 2.02; 95%CI = 1.77–2.31). Parents’ unemployment (one or both parents) was associated with the dimensions “school problems” (OR = 1.38; 95%CI = 1.20–1.58), “death/severe disease” (OR = 1.15; 95%CI = 1.00–1.32), and “household dysfunction” (OR = 1.86; 95%CI = 1.61–2.16).

Table 3 shows the association between the different dimensions of ACEs and the child’s health status and behaviors. Increased odds of any disease diagnosis (AOR = 1.35; 95%CI = 1.06–1.72), asthma diagnosis (AOR = 1.30; 95%CI = 1.04–1.62), and excess of screen activities (AOR = 1.15; 95%CI = 1.00–1.32) were observed in children reporting “death/severe disease”, after adjustment for sex and household income. “Life changes” were also associated with an increased odds of asthma diagnosis (AOR = 1.22; 95%CI = 1.00–1.50) and excess screen activities (>480 min/week) (AOR = 1.24; 95%CI = 1.09–1.41). “Household dysfunction” also increased the odds of excess screen activities (AOR = 1.22; 95%CI = 1.04–1.44).

By analyzing the cumulative effect of ACEs, using the sum of the different dimensions, we observed an increased risk of excess screen activities, after adjustment for sex and household income, in children reporting two or more dimensions of ACEs (Table 4).

## 4. Discussion

The present study involving a large population-based birth cohort provided evidence that ACE dimensions at 10 years of age were associated with sociodemographic characteristics, child characteristics, and health-related behaviors.

The prevalence of exposure to ACEs among 10-year-old children was high, with most children (96.2%) reporting having been exposed to at least one ACE. The most prevalent types of ACEs reported were a household member shouting, yelling, or screaming at the child, witnessing parents arguing or fighting, the death of someone close to the child, and ever being beaten and hurt at school. High prevalence of ACEs is also observed in other Portuguese samples, although in adults, e.g., 88% in morbid obesity candidates for bariatric surgery [5] and 68.2% of at least one ACE before the age of 18 reported during adult life [6]. Lower values are observed in young adults between 18 and 30 years, reporting 24% of ACEs [7]. Moreover, even though any type of maltreatment against children was fully prohibited by law in Portugal in 2007 [48], having a household member hitting, kicking, or punching was reported by almost one-fifth of Generation XXI participants. The results observed were similar to the 22.6% for lifetime prevalence of physical abuse, mostly perpetrated at home, before the age of 18 years reported by the WHO [49] and about half of them reported among Taiwanese children [50]. The prevalence of physical abuse observed in Generation XXI children was also different from the one reported in retrospective studies in Portuguese adults (6.7%) [51] and in adults from the United Kingdom (7.6%) [52]. These differences might be explained by methodological choices or even by the timing of the report. In our study, children reported experiences that occurred in their lives as they grew up. The previously reported studies in Portugal and the United Kingdom have data retrospectively collected in adulthood, and thus, some experiences might not be recalled and consequently not reported.

We observed that higher maternal and paternal education increased the likelihood of “abuse” and “life changes”, while it decreased the likelihood of “school problems” and “household dysfunction”. We hypothesized that as “abuse” included items such as parents fighting, screaming, and swearing at the child, and beating the child, the weight of items associated with emotional abuse (all statistically and positively associated with both maternal and paternal education) might contribute to the obtained result. Nevertheless, this is a result that is not supported by the literature on child rearing practices, which states that parents with lower education levels use harsher practices [53,54]. Lower levels of formal education are usually associated with a number of stress factors that make up the daily lives of families and promote the use of more coercive disciplinary strategies [55]. However, as our dimension of abuse includes items that are not physical, children from highly educated parents are more likely to report and to value the parents fighting, screaming, and swearing at the child, and beating the child. There are studies showing that high socioeconomic participants are more likely to report emotional or psychological abuse when compared to their counterparts [56], and the same seems to be observed in our sample. Regarding the item “life changes”, the tendency was for these changes to occur in more socioeconomically advantaged environments, where people were more capable of living separated. Lower household income increased the odds of the child suffering from “abuse”, “school problems”, “death/severe disease”, “life changes”, and “household dysfunction”. Moreover, having both parents unemployed increased the likelihood of having “school problems” and “household dysfunction”. In accordance, recent research had shown that ACEs were strongly socioeconomically patterned at both the family [57] and area level [58]. Thus, research has been calling attention to the need for focusing on poverty and socioeconomic inequality as a cause of ACEs [57,59,60].

The association between different dimensions of ACEs and the child’s health status and child behaviors was also observed. Results showed that after adjustment for sex, increased odds of asthma or any other disease diagnosis were observed in children reporting “death/severe disease”. “Life changes” were also associated with an increased risk of asthma diagnosis. These results were aligned with a previous systematic review that also observed that exposure to traumatic stress in childhood significantly increased the risk of asthma onset [12], and this may be explained by the process of biological embedding of stress or body programing that may lead to later disease development.

“Life changes”, “school problems”, and “household dysfunction” increased the risk of children spending more time on screen activities. Similar results were observed in the CDC-Kaiser ACE studies, where when compared to participants who had no ACEs, those who had experienced four or more categories of childhood exposure had a 1.4-fold higher likelihood of physical inactivity [1]. Additionally, similar results were reported in a WHO report, which conducted ACEs surveys in eight eastern European countries, and observed that young adults who reported at least four ACEs were at increased risk of many health-harming behaviors, including physical inactivity, when compared with those who reported no ACEs [9].

Previous studies had shown that ACEs usually co-occur and may have a cumulative effect [16,17,18]. Additionally, in CDC-Kaiser ACE studies, it was observed that child maltreatment and adverse household characteristics were co-occurring phenomena, so the presence of one ACE predicted the odds of exposure to additional ACEs [61]. Therefore, we analyzed the cumulative effect of the different ACEs and how they may have affected children’s health. Additionally, we observed that the increased number of ACEs dimensions did not increase the odds of any disease diagnosis, asthma diagnosis, obesity, or low consumption of fruits and vegetables. However, it increased the odds of excess screen activities almost 2-fold. However, when we look at the different dimensions, “school problems” and “death/severe disease” increased the odds of any disease diagnosis; “death/severe disease”, “life changes”, and “household dysfunction” increased the risk of asthma diagnosis; children in the “school problems” dimension were at increased risk of obesity, and all dimensions were associated with increased risk of excess of screen activities. These results seem to show that different exposures might be associated with different outcomes in health and behaviors. For example, “abuse” and “school problems” might represent different entities of abuse, which might have distinct effects on disease onset or obesity. Furthermore, as the frequency of the different types of ACEs in every individual was not assessed, it could be assumed that in some types of traumas, such as death or severe disease in the family, one single event might already have substantial consequences, whereas other types, such as life changes, might require longer-lasting adverse conditions.

As our study focused on children, the time of exposure may not be enough to impact physiological systems, but an embodiment process of adversity may already be on course. Additionally, it is not expected that experiencing adversity will lead to an immediate onset of disease in childhood. However, the underlying process might have a long asymptomatic phase of development that may start during early childhood.

### Limitations and Strengths

This study has some strengths and limitations that should be acknowledged. The main strength was that the assessment of ACEs was conducted within the scope of a large sample of 10-year-old children participating in the Portuguese population-based birth-cohort, the Generation XXI. ACEs were directly reported by school-aged children in a confidential and trustworthy environment, comprising a broad range of items from the household and familiar environment. According to the literature, the ACE questionnaire is a reliable, valid, and economic screen for the retrospective assessment of adverse childhood experiences [62], showing acceptable to high internal consistency, test–retest reliability [63], concurrent and convergent validity, and findings replicated across samples [62]. However, it has been described that the scale presents some problems, which included: (1) limited item coverage, (2) collapsing of items and response options, (3) simplistic scoring approach, and (4) lack of psychometric assessment. While epidemiological research on ACEs may be useful evidence for population-level or structural policies, it might be an insufficient and ill-adapted tool for implementation by social or healthcare workers [33,36,64].

There is also the potential for under-reporting, which should not be disregarded, as children may not have a level of maturity and understanding to recognize some of the experiences. Sexual abuse was not included in our assessment, although it is known to be one of the most traumatic events. As the evaluation of ACEs was conducted through a self-completed questionnaire, and sexual abuse may be a hypersensitive question, the cohort’s coordination team decided not to pose it. However, we would expect similar results for the sexual abuse item, although we cannot confirm whether sexual abuse would be considered in each dimension or whether it would be posed as a separate dimension. This is expected, namely, because sexual abuse usually co-occurs with other adversities [16,17,18] and also because the mechanisms by which the exposure to sexual abuse leads to health consequences and poor health behaviors are also related to damage of the stress-response systems, like any other adverse childhood experience [34].

The use of principal component analysis allowed the study of the original items of ACEs in groups of correlated ACEs and reduced the size of the dataset by identifying the underlying dimensions within the data [39]. The number of dimensions that best represents the data was chosen based on the scree plot [40], the size of the factor loadings, and the interpretability of the resulting dimensions. The labels to define dimensions might not be perfect but were defined to be as close as possible and to facilitate the reporting of the results. Although we combined information across 15 different domains to characterize a wide range of childhood adversity, our approach did not account for the severity of different types of events possibly captured by other weighting schemes. Moreover, even though we cannot distinguish between the different dimensions when looking at the cumulative effect, when more than two dimensions are present, independent of which ones, the health status of the child seems to worsen.

Thus, the results can be presented by the different dimensions that reflect different types of ACEs and using a cumulative effect of different dimensions, including several ACEs. A strength of this analysis is that each dimension was distinguishable by the item of ACEs loading on it. As a limitation of the questionnaire, the timing or the frequency of adversity were not accounted for, and it could be that repeated exposure to events deemed as lower intensity could be more strongly related to adverse outcomes than an isolated event rated as highly intense. However, we believe that the self-report of ACEs in the first decade of life accounts for experiences that had an impact on the child’s life, even if some children may not have an emotional reaction or recognition of their ACEs.

As observed before [65], Cronbach’s alpha values were low for the majority of the dimensions. However, it should be noted that, globally, the alpha values may not truly reflect the internal consistency of a measure, and thus, the structure of the questionnaire should not be disregarded in light of the correlations between the items within each dimension [66,67]. Indeed, given the formative nature of PCA, Cronbach’s alpha values may not be meaningful for PCA dimensions, given that it is not expected that dimensions necessarily correlate and exhibit internal consistency, unlike reflective models [68]. Moreover, Cronbach’s alpha values tend to be lower for both binary data and when using formative modeling techniques [69]. Finally, Cronbach’s alpha does not perform well when dimensions comprise a few items (e.g., the life changes dimension comprises two items [67]). Therefore, the dimensions yielded by the PCAs should not be disregarded based on Cronbach’s alpha values alone.

The present results indicated that when 15 items are used in a PCA, a five-dimensional structure is best suited to the data, though this approach accounts for minimal amounts of the overall variance. Although replication is needed, these five dimensions may most accurately reflect the types of ACEs, setting the stage for future work to utilize these dimensions as correlates of myriad outcomes, including physical and mental health in 10-year-old children. More globally, the present findings highlight the importance of selecting measurement strategies that are both theoretically and mathematically grounded to establish valid and reliable assessment tools. This is paramount in fostering the researcher’s ability to understand true relations between ACEs and outcomes of interest that will be assessed in future cohort waves.

## 5. Conclusions

In summary, ACEs dimensions at 10 years of age are associated with sociodemographic characteristics. These data illustrate the potential social patterning of ACEs.

The focus should be on the development and evaluation of programs that prevent the occurrence of childhood adversities in the first place and then the experimental demonstration of the population health effects from their dissemination. Our findings highlight the need for healthcare professionals to assess adversity exposure throughout childhood to identify children at risk of developing harmful health-related behaviors and consequent illness, thus stressing the opportunity for effective early intervention to limit or ameliorate the impact of violence and other adverse experiences across the lifespan.

Further research is required to measure ACEs dimensions in different populations and age ranges—in particular, the replication of these results in different childhood populations and confirmatory analysis of the five dimensions that most accurately reflect the types of ACEs, setting the stage for future work to utilize these dimensions as correlates of different health outcomes. Additionally, future studies should focus on other ACEs, such as sexual abuse or displacement caused by natural disasters (e.g., hurricanes) or man-made disasters (e.g., war), to determine if individuals who experienced these specific types of ACEs would be at a higher risk of other illnesses. ACEs can be prevented, and therefore, more investment in policies and programs that effectively improve child well-being should be a priority. Moreover, research is still needed to better understand the mechanisms explaining the emergence and persistence of poorer health outcomes later in life in victims of abuse. Thus, efforts focusing on preventing, identifying, and stopping ACEs exposure should be in place to better protect children. When prevention of victimization is no longer possible, efforts should be made to mitigate the health consequences.

## Figures and Tables

**Table 1 ijerph-19-08344-t001:** Prevalence of adverse childhood experiences (ACEs) at 10 years of age according to the child’s sex, and eigenvector values for the five dimensions of ACEs in Generation XXI (*n* = 5295).

	Prevalence (%)			Dimensions of ACEs				
	All (*n* = 5295)	Girls (*n* = 2598)	Boys (*n* = 2697)	Abuse	School Problems	Death/ Severe Disease	Life Changes	Household Dysfunction
				α = 0.332	α = 0.361	α = 0.403	α = 0.493	α = 0.397
Did your parents ever separate or divorce?	20.6	20.4	20.8	−0.034	0.128	−0.082	**0.658**	0.054
Did you ever move from a house, school, or neighborhood?	42.0	42.5	43.0	0.081	−0.116	0.099	**0.648**	−0.074
Have you ever had difficulties at school?	40.0	39.5	40.4	−0.026	**0.513**	0.116	−0.209	0.059
Have you ever witnessed your parents arguing or fighting?	44.4	40.1	48.6	**0.306**	0.044	0.091	−0.009	0.234
Have you ever heard your parents talking about the household financial hardships?	22.4	22.1	22.7	0.137	−0.056	0.202	−0.118	**0.479**
Did someone you were very close to (family or friend) die?	43.5	43.2	43.7	−0.065	−0.034	**0.673**	−0.022	−0.088
Did someone in the household have an injury or severe illness?	34.8	33.2	36.4	0.070	−0.010	**0.562**	0.051	−0.014
Has someone in your school ever beaten and hurt you?	42.8	35.1	50.3	0.232	**0.503**	−0.075	0.001	−0.078
Is someone in the household a problem drinker or uses street drugs?	1.6	1.1	2.1	−0.037	−0.021	−0.142	−0.005	**0.723**
Did a household member ever go to prison?	3.6	3.7	3.4	−0.161	0.075	0.082	0.247	**0.395**
Did you have an illness or accident that forced you to stay or go to the hospital many times?	12.9	10.1	15.5	−0.040	0.162	**0.333**	0.052	0.028
Did someone in the household shout, yell, or scream at you?	57.7	52.0	63.1	**0.448**	0.063	0.045	−0.060	0.060
Did someone in the household swear at, insult, put you down, or humiliate you?	9.5	7.2	11.7	**0.528**	−0.073	−0.037	0.062	−0.035
Did someone in the household hit, kick, or punch you?	18.5	16.4	20.4	**0.546**	−0.010	−0.049	0.046	−0.064
Were your parents ever called to the school because you did something wrong?	12.8	6.2	19.2	−0.100	**0.637**	−0.051	0.114	−0.030

Bold for grouping items in different dimensions.

**Table 2 ijerph-19-08344-t002:** Descriptive characteristics of sample according to the five ACEs’ dimensions from principal component analysis.

		Abuse *n* (%)	95%CI	School Problems *n* (%)	95%CI	Death/Severe Disease *n* (%)	95%CI	Life Changes *n* (%)	95%CI	Household Dysfunction *n* (%)	95%CI
		3786 (71.5)		3349 (63.2)		3361 (63.5)		2698 (51.0)		1354 (25.6)	
**Sociodemographic characteristics**											
Family structure	Both parents	2909 (70.3)	68.9–71.7	2575 (62.2)	60.8–63.8	2575 (62.2)	60.8–63.7	1641 (39.7)	38.2–41.1	1019 (24.6)	23.4–26.0
	Only mother/only father	837 (76.0)	73.4–78.4	736 (66.9)	64.1–69.6	745 (67.8)	65.1–70.5	1010 (91.9)	90.0–93.3	303 (27.6)	25.0–30.2
	Neither mother nor father	33 (66.7)	52.8–78.2	33 (67.2)	53.9–78.3	36 (72.0)	58.9–82.5	41 (82.0)	68.4–89.5	28 (56.0)	42.1–68.6
Maternal education	≤9th grade	1336 (69.5)	67.3–71.5	1326 (69.0)	67.0–71.1	1229 (64.1)	61.9–66.2	911 (47.4)	45.1–49.5	566 (29.4)	27.5–31.6
	10th–12th grade	1153 (71.9)	69.7–74.1	1027 (64.1)	61.8–66.4	994 (62.0)	59.5–64.3	798 (49.8)	47.3–52.2	422 (26.3)	24.2–28.5
	>12th grade	1179 (73.2)	71.0–75.3	891 (55.3)	52.9–57.7	1022 (63.5)	61.1–65.8	853 (53.0)	50.1–55.6	303 (18.8)	17.0–20.8
Paternal education	≤9th grade	1424 (69.0)	67.0–71.0	1407 (68.2)	66.2–70.2	1322 (64.1)	62.0–66.1	791 (38.3)	36.2–40.4	597 (28.9)	27.1–31.1
	10th–12th grade	860 (70.8)	68.2–73.3	717 (59.0)	56.3–61.8	750 (61.7)	58.9–64.4	516 (42.5)	39.7–45.3	281 (23.1)	20.9–25.7
	>12th grade	706 (73.5)	70.6–76.2	523 (54.4)	51.3–57.6	581 (60.5)	57.3–63.5	434 (45.2)	42.2–48.4	170 (17.7)	15.3–20.1
Income	<EUR 1000	1006 (73.4)	70.9–75.6	974 (71.0)	68.5–73.3	910 (66.4)	64.0–68.9	829 (60.5)	57.8–63.0	494 (36.0)	33.5–38.6
	EUR 1001–2000	1706 (69.9)	68.1–71.2	1561 (64.0)	62.2–66.0	1506 (61.7)	59.8–63.6	1149 (47.1)	45.1–49.1	576 (23.6)	22.1–25.5
	>EUR 2000	946 (73.0)	70.6–75.4	696 (53.7)	0.51–56.5	827 (63.8)	61.1–66.3	607 (46.8)	44.3–49.7	235 (18.1)	16.1–20.2
History of parental unemployment	None of the parents	1971 (69.5)	67.9–71.2	1697 (59.9)	58.1–61.7	1735 (61.2)	59.4–63.0	1116 (39.4)	37.6–41.2	589 (20.8)	19.3–22.3
	One of the parents	816 (72.3)	69.6–74.8	750 (66.4)	63.6–69.1	730 (64.6)	61.8–67.4	463 (41.0)	38.1–43.8	356 (31.5)	29.0–34.4
	Both parents	119 (72.1)	64.8–78.4	120 (73.2)	66.1–79.5	104 (63.6)	56.0–70.6	71 (43.0)	35.7–50.7	67 (40.9)	33.6–48.5
**Child characteristics and health-related behaviors**											
Sex	Girl	1744 (67.1)	65.3–68.9	1476 (56.8)	55.0–58.8	1602 (61.7)	59.7–63.4	1305 (50.2)	48.3–52.2	656 (25.3)	23.7–27.0
	Boy	2042 (75.7)	74.0–77.3	1873 (69.5)	67.8–71.2	1759 (65.2)	63.5–67.1	1393 (51.7)	49.8–53.6	698 (25.9)	24.3–27.6
Low consumption of fruits and vegetables	Yes	2462 (71.2)	69.6–72.6	2185 (63.2)	61.6–64.8	2189 (63.3)	61.7–64.9	1746 (50.5)	48.9–52.1	853 (24.7)	23.3–26.2
	No	1285 (72.4)	70.3–74.5	1125 (63.5)	61.2–65.7	1134 (63.9)	61.7–66.2	914 (51.5)	49.2–53.8	486 (27.4)	25.4–29.5
Excess screen activities	Yes	970 (75.2)	72.8–77.5	864 (67.0)	64.4–69.5	850 (65.9)	63.2–68.4	714 (55.4)	52.5–57.9	366 (28.4)	26.0–30.9
	No	2816 (70.3)	68.9–71.7	2485 (62.1)	60.6–63.6	2511 (62.7)	61.3–68.4	1984 (49.5)	48.1–51.2	988 (24.7)	23.4–26.0
**Child health status**											
Any disease diagnosis	Yes	246 (73.4)	68.2–77.7	228 (68.2)	63.1–73.0	233 (69.6)	64.5–74.3	181 (54.0)	48.9–59.6	96 (28.7)	23.9–33.5
	No	3525 (71.4)	70.1–72.6	3109 (62.9)	61.6–64.3	3116 (63.1)	61.8–64.4	2507 (50.8)	49.4–52.2	1250 (25.3)	24.2–26.6
Asthma diagnosis	Yes	302 (71.8)	67.3–75.9	272 (64.8)	60.5–69.5	291 (69.3)	64.4–73.2	235 (56.0)	51.4–60.9	119 (28.3)	24.5–33.1
	No	3452 (71.4)	70.1–72.7	3052 (63.2)	61.8–64.5	3041 (62.9)	61.6–64.4	2441 (50.5)	49.1–51.9	1222 (25.3)	24.1–26.5
BMI	Underweight	46 (75.4)	63.1–84.6	42 (68.9)	56.2–79.2	34 (55.7)	43.2–67.6	26 (42.6)	30.9–55.2	13 (21.3)	12.8–33.3
	Normal	2136 (71.2)	69.5–72.8	1854 (61.8)	60.1–63.5	1905 (63.5)	61.8–65.2	1530 (51.0)	49.2–52.8	729 (24.3)	22.8–25.9
	Overweight	983 (72.2)	69.8–74.6	864 (63.5)	61.0–66.1	864 (63.6)	61.0–66.1	695 (51.1)	48.5–53.8	362 (26.6)	24.3–29.0
	Obese	621 (71.2)	68.1–74.1	588 (67.4)	64.4–70.7	558 (64.0)	60.8–67.2	447 (51.3)	47.8–54.4	250 (28.7)	25.8–31.8

**Table 3 ijerph-19-08344-t003:** Associations (Odds Ratio and 95% Confidence Interval, OR (95%CI)) of child health status and behaviors with each dimension of adverse childhood experiences (ACEs) in the Generation XXI (*n* = 5295).

			Abuse		School Problems		Death/Severe Disease		Life Changes		Household Dysfunction	
			OR (95% CI)	AOR (95% CI)	OR (95% CI)	AOR (95% CI)	OR (95% CI)	AOR (95% CI)	OR (95% CI)	AOR (95% CI)	OR (95% CI)	AOR (95% CI)
**Child health status**	**Disease diagnosis** **(reference: no)**	**Any disease**	1.10 (0.85–1.41)	1.11 (0.86–1.43)	1.26 (1.00–1.60)	1.30 (1.00–1.69)	1.34 (1.05–1.70)	1.35 (1.06–1.72)	1.15 (0.92–1.44)	1.12 (0.89–1.40)	1.17 (0.92–1.50)	1.12 (0.89–1.40)
		**Asthma**	1.02 (0.82–1.27)	0.98 (0.78–1.23)	1.09 (0.89–1.34)	1.06 (0.85–1.32)	1.30 (1.05–1.61)	1.30 (1.04–1.62)	1.26 (1.03–1.54)	1.22 (1.00–1.50)	1.18 (0.95–1.47)	1.22 (1.00–1.50)
	**Obesity (BMI > +2 SD)**		0.96 (0.81–1.13)	0.91 (0.76–1.10)	1.26 (1.08–1.47)	1.22 (1.04–1.44)	1.02 (0.88–1.19)	0.99 (0.85–1.16)	1.00 (0.87–1.16)	0.96 (0.83–1.12)	1.20 (1.02–1.42)	0.96 (0.83–1.12)
	**BMI** (reference: underweight)	**Normal**	0.76 (0.45–1.45)	0.77 (0.42–1.41)	0.73 (0.42–1.27)	0.78 (0.45–1.36)	1.38 (0.83–2.30)	1.36 (0.81–2.28)	1.40 (0.84–2.35)	1.41 (0.84–2.37)	1.19 (0.64–2.20)	1.41 (0.84–2.37)
		**Overweight**	0.85 (0.47–1.54)	0.81 (0.44–1.49)	0.79 (0.45–1.37)	0.83 (0.48–1.46)	1.39 (0.83–2.32)	1.36 (0.81–2.30)	1.41 (0.84–2.37)	1.42 (0.84–2.41)	1.34 (0.72–2.50)	1.42 (0.84–2.40)
		**Obese**	0.81 (0.44–1.47)	0.75 (0.40–1.39)	0.95 (0.54–1.66)	0.98 (0.55–1.73)	1.42 (0.84–2.39)	1.35 (0.79–2.29)	1.41 (0.83–2.38)	1.36 (0.80–2.31)	1.49 (0.79–2.79)	1.36 (0.80–2.31)
**Child behaviors**	**Low consumption of fruits and vegetables**		1.06 (0.94–1.21)	1.04 (0.91–1.19)	1.01 (0.90–1.14)	0.97 (0.85–1.09)	1.03 (0.91–1.16)	1.01 (0.89–1.14)	1.04 (0.93–1.17)	1.01 (0.89–1.13)	1.95 (1.31–2.91)	1.01 (0.89–1.13)
	**Excess screen activities**		1.28 (1.11–1.48)	1.27 (1.10–1.48)	1.24 (1.09–1.41)	1.22 (1.07–1.41)	1.14 (1.99–1.30)	1.15 (1.00–1.32)	1.25 (1.11–1.42)	1.24 (1.09–1.41)	2.31 (1.49–3.58)	1.24 (1.09–1.41)

AOR: Adjusted Odds Ratio for sex and household income.

**Table 4 ijerph-19-08344-t004:** Associations (Odds Ratio and 95% Confidence Interval, OR (95%CI)) of child health status and behaviors with the sum of the dimensions of adverse childhood experiences (ACEs) in the Generation XXI (*n* = 5295).

			0 Dimensions	1 Dimension		2 Dimensions		3 Dimensions		4 Dimensions		5 Dimensions	
				OR (95% CI)	AOR (95% CI)	OR (95% CI)	AOR (95% CI)	OR (95% CI)	AOR (95% CI)	OR (95% CI)	AOR (95% CI)	OR (95% CI)	AOR (95% CI)
**Child health status**	**Disease** **diagnosis** **(reference: no)**	**Any disease**	Reference	0.74 (0.39–1.41)	0.76 (0.39–1.47)	0.65 (0.36–1.18)	0.68 (0.34–1.26)	0.98 (0.55–1.74)	1.97 (0.53–1.76)	1.11 (0.62–1.99)	1.11 (0.61–2.04)	1.22 (0.64–2.36)	1.23 (0.62–2.43)
		**Asthma**	Reference	0.88 (0.48–1.62)	0.92 (0.48–1.75)	0.96 (0.54–1.69)	1.04 (0.57–1.90)	1.08 (0.62–1.89)	1.12 (0.61–2.02)	1.21 (0.69–2.13)	1.23 (0.67–2.25)	1.54 (0.83–2.86)	1.56 (0.81–3.01)
	**Obesity (BMI > +2 SD)**		Reference	1.14 (0.74–1.77)	1.06 (0.68–1.67)	1.01 (0.66–1.53)	0.94 (0.61–1.44)	1.13 (0.75–1.71)	1.02 (0.67–1.55)	1.14 (0.87–1.99)	1.15 (0.75–1.76)	1.26 (0.79–2.01)	1.06 (0.66–1.73)
**Child** **behaviors**	**Low consumption of fruits and vegetables**		Reference	1.25 (0.89–1.77)	1.19 (0.84–1.70)	1.23 (0.89–1.70)	1.17 (0.84–1.70)	1.21 (0.88–1.67)	1.13 (0.81–1.56)	1.29 (0.93–1.78)	1.17 (0.84–1.63)	1.46 (1.01–2.11)	1.26 (0.86–1.85)
	**Excess screen activities**		Reference	1.36 (0.82–1.92)	1.35 (0.86–2.11)	1.55 (1.04–2.31)	1.63 (1.07–2.48)	1.89 (1.27–2.80)	2.03 (1.33–3.08)	1.95 (1.31–2.91)	2.06 (1.35–3.14)	2.31 (1.49–3.58)	2.35 (1.48–3.75)

AOR: Adjusted Odds Ratio for sex and household income.

## Data Availability

Data is available from Cohort coordination upon request.

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
