# Peer review of "Prevalence of Adverse Childhood Experiences in the First Decade of Life: A Study in the Portuguese Cohort, Generation XXI"

_ijerph, 2022, doi:10.3390/ijerph19148344_

Round 1

Reviewer 1 Report

Dear authors, thank you for giving me the opportunity to review your manuscript: “Prevalence of adverse childhood experiences in the first decade of life: a study in the Generation XXI cohort”. 

I think that this manuscript can meaningfully contribute to the literature. 

I send below some comments to improve the manuscript.

Title:

-        I think the title should have reference to the Portuguese population.

Introduction:

-        There are several Portuguese studies that study ACEs. It would be an added value to include some of the results of those studies in the introduction to understand the results of that country at the ACEs level (population studied, the impact, prevalence).

Materials and Methods: 

-        Please provide details about the inclusion criterium. 

-        Please include how long it took for the protocol to be answered by the participants.

-        “ACEs items with loadings >0.300 on a dimension, were considered to have a strong association with the dimension and were deemed to be the most informative in describing it.” Authors should provide a reference to sustain the choice to consider items with loadings >0.300.

-        On ACEs dimensions derived by principal component analysisplease provide the Cronbach alpha of each dimension (childhood abuse; problems at school; death or severe disease in the family; life changes; and household dysfunction).

Discussion

-        Authors should explain the results “The prevalence of physical abuse observed in Generation XXI children was also different from the one reported in retrospective studies in Portuguese adults (6.7%) [45].”. 

o   Concerning the Portuguese studies, why is there such a difference? 

o   There are more Portuguese studies evaluating physical abuse of Portuguese youth that identify values more like those found by the authors of this article. I think the authors should do a broader survey that addresses these studies.

-        Authors wrote: “We hypothesized that as “childhood abuse” included items such as parents fighting, screaming, and swearing at the child, and beating the child, the weight of items associated with emotional abuse (all statistically and positively associated with both maternal and paternal education) might contribute to the obtained result.” Authors should explain why the results were obtained. Why does paternal education increase the likelihood of fighting, screaming, swearing at the child, and beating the child?

-        This sentence seems to be out of context: “Additionally, advances in technology have led to a marked increase in screen time and sedentary behaviour among adolescents, related to lower energy expenditure [51].” Please provide a rationale for this sentence.

-        Authors only present some results without discussing them. Please discuss this result presented in the discussion: “Therefore, we analysed the cumulative effect of the different ACEs and how they may have affected children’s health. Although we cannot discard the suffering of the children when these events occurred, and their effect on their wellbeing, the exposure to each ACEs individually seemed to have a limited impact on children’s health as it does not seem to have impacted disease diagnosis or BMI.”

Conclusion

-        Authors should add future directions.

This manuscript has a lot to offer, but it needs revisions to expand the literature review and discuss some results presented.

Reviewer 2 Report

This manuscript describes the prevalence of Adverse Childhood Experiences (ACE) evaluated according to children’s perspective at 10-years of age, as well as the association of this evaluation with sociodemographic characteristics and children’s health and behaviors. This study is part of a large ongoing cohort study in the Porto Metropolitan Area in Portugal.

The study is conceptually well-supported, easy to read and well organized. Using children as informants of their ACE is the main argument used by authors to justify the study relevance and novelty, which I agree. However, some specific methodological aspects deserve clarification before the manuscript can be recommended for publication.

1)      Some more detailed information about how the ACE information was collected is missing. For instance, what instruction was given to children before filling the questionnaire? Was it in an individual or group format setting?

2)      A Principal Component Analysis (PCA) on the 15 items of the ACE questionnaire was conducted to identify underlying dimensions. Given the amount of evidence already available about ACE, I would expect to find a less exploratory analytic approach and a more conceptually grounded Confirmatory Factor Analysis.

3)      One of the dimensions identified by the PCA comprises only two items (Life Changes), which is psychometrically not recommended for identifying dimensions.

4)      I couldn’t understand to what refers the n presented on Table 2, because it is unclear how each PCA dimension score was calculated. Is it presence/absence? If this is the case, I would suggest using a score derived from the sum of the number of “yes responses” in each dimension. This score would range from 0 to the number of items in the dimension. Using this kind of score would allow authors to conduct an analysis of variance in further analyses.

5)      Authors should use p-values with caution because of the magnitude of the sample size, very small differences are statistically significant. Other kind of effect sizes should be used.

6)      Table 4 is missing in the manuscript, so I can’t appreciate the results regarding the cumulative effects.

7)      Some minor suggestions regarding tables are also relevant. Tables are not reader-friendly. They should be edited, the position occupied by the ACE dimensions variable should always be the same and the dimensions should be presented by a shorter name.

8)      From my perspective, Figure 1 is not necessary for the understanding of the information presented.

 I hope the authors find my comments useful.

Round 2

Reviewer 1 Report

The authors have made an effort to answer all the questions asked. However, the instrument used should have been validated for the Portuguese population.